# Improving MRI-based Knee Disorder Diagnosis with Pyramidal Feature Details

**Matteo Dunnhofer**[1]    MATTEO.DUNNHOFER@UNIUD.IT
**Niki Martinel**[1]    NIKI.MARTINEL@UNIUD.IT
**Christian Micheloni**[1]    CHRISTIAN.MICHELONI@UNIUD.IT
[1] *Machine Learning and Perception Lab, University of Udine, Udine, Italy*

## Abstract

This paper presents MRPyrNet, a new convolutional neural network (CNN) architecture that improves the capabilities of CNN-based pipelines for knee injury detection via magnetic resonance imaging (MRI). Existing works showed that anomalies are localized in small-sized knee regions that appear in particular areas of MRI scans. Based on such facts, MRPyrNet exploits a Feature Pyramid Network to enhance small appearing features and Pyramidal Detail Pooling to capture such relevant information in a robust way. Experimental results on two publicly available datasets demonstrate that MRPyrNet improves the ACL tear and meniscal tear diagnosis capabilities of two state-of-the-art methodologies. Code is available at https://git.io/JtMPH.

**Keywords:** MRI, Knee Disorder Diagnosis, ACL Tear Detection, Meniscus Tear Detection, Feature Pyramid Network, Pyramidal Detail Pooling.

## 1. Introduction

Due to its complex and delicate anatomy, the knee is one of the body regions most susceptible to severe injuries. To evaluate and diagnose such disorders, magnetic resonance imaging (MRI) is currently the standard clinical modality. This is because of the high accuracy this sensory system gives for the diagnosis of pathologies such as the anterior cruciate ligament (ACL) tear and meniscus tear (Rangger et al., 1996; Mackenzie et al., 1996; Cheung et al., 1997; Oei et al., 2003). Indeed, the efficacy of MRI makes the knee the body structure where musculoskeletal (MSK) examinations are performed more. However, an accurate interpretation of knee MRIs requires a significant effort because of the volume and detail of information contained in MRI exams. As shown by Kim et al. (2008), such a manual process can lead to improper outcomes even when performed by board-certified MSK radiologists. To mitigate these problems, Bien et al. (2018) recently proposed MRNet, a fully automated system for knee MRI interpretation based on convolutional neural networks (CNNs). The solution presents a pipeline in which an AlexNet-based CNN architecture (Krizhevsky et al., 2012) is trained to predict the presence or absence of ACL tear, meniscal tear, or generic abnormalities, given axial, coronal, and sagittal views of an MRI exam. For the same tasks, Tsai et al. (2020) later proposed ELNet as an improved CNN architecture based on residual connections, normalization layers, and blur pooling. Azcona et al. (2020) instead studied the impact of transfer learning and data augmentation techniques on deep residual models.

Despite the promising results, the aforementioned deep learning solutions neglected the particular anatomy of knee disorders. Indeed, the currently available methods learn an

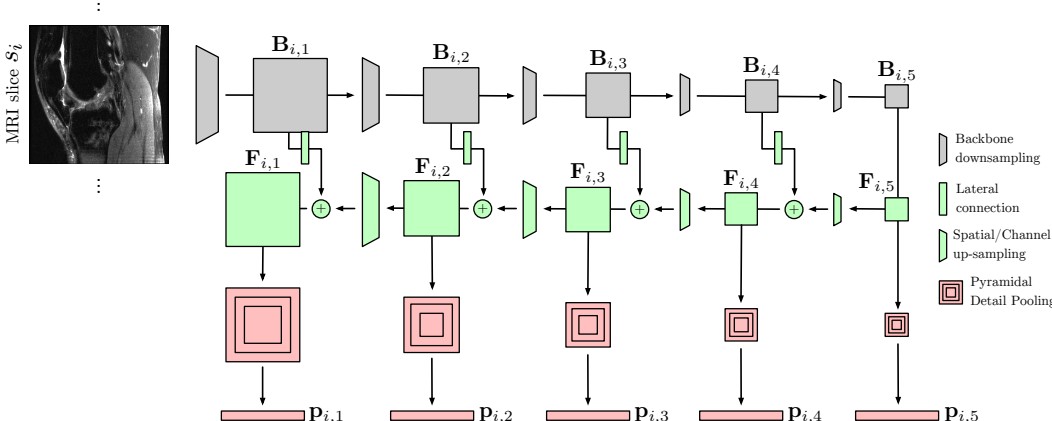

Figure 1: Processing procedure of MRPyrNet. Each MRI slice $s_i$ passes through a backbone CNN (represented by the gray shapes). The feature maps produced are exploited in an FPN strategy (represented by the green shapes), and in turn analyzed by the PDP modules (light red shapes). These build the vectors $\{\mathbf{p}_{i,l}\}_{l=1}^{L}$ which consist of representations containing MRI information at multiple levels of detail.

MRI image representation that ultimately compresses, by a global pooling operation, the 2D spatial information of CNN feature maps into a vector of single scalar values. We argue that such a process harms the detection of the relevant features of anomalies and that it ultimately leads to wrong diagnosis predictions in more difficult and ambiguous cases. Several works (Hash, 2013; Nguyen et al., 2014; Štajduhar et al., 2017; Lecouvet et al., 2018; Bien et al., 2018) have shown that particular features associated with the diagnoses are localized in specific regions of the knee, that appear very small and often localized around the center of the MRI images. This is especially true for the ACL tear (Hash, 2013) and meniscal tear (Nguyen et al., 2014; Lecouvet et al., 2018).

Following such an intuition, we present MRPyrNet, a novel CNN architecture capable of capturing the relevant information of knee disorders in such image areas. Since feature pyramids (Adelson et al., 1984) showed to be effective for the better detection of pathologies (Liu et al., 2019b; Xiao et al., 2019; Hao et al., 2020), our solution comprises a Feature Pyramid Network (FPN) (Lin et al., 2017) to better detect small-sized features, and a Pyramidal Detail Pooling (PDP) module to analyze features at multiple levels of detail. Our plug-and-play strategy can be applied to any existing backbone CNN. We show how to apply it to the pipelines of MRNet (Bien et al., 2018) and ELNet (Tsai et al., 2020) for the detection of the ACL and meniscal tears. Our experiments indicate that including priors on the disorders is important to improve the diagnostic capabilities of state-of-the-art methodologies.

## 2. Methodology

In this section, we describe the processing procedure of MRPyrNet and of its components, which are shown as a whole in Figure 1. We assume an MRI exam consisting of a sequence

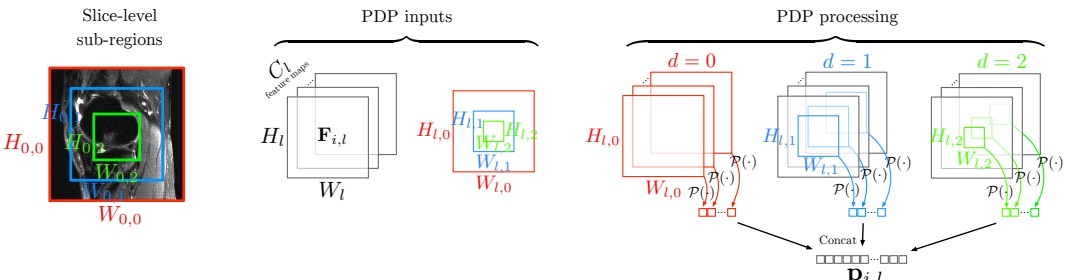

Figure 2: Visual representation of the operation performed by the PDP module with $D = 3$. PDP gets in input the FPN feature maps $\mathbf{F}_{i,l}$, and a set of $D$ sub-regions, located in $\mathbf{F}_{i,l}$ center, with sizes $(W_{l,d}, H_{l,d})$ that are obtained after the dimensions $(W_{0,d}, H_{0,d})$ generated at slice level. For each detail level $d$, $\mathbf{F}_{i,l}$ is pooled channel-wise by means of the function $\mathcal{P}(\cdot)$ in the area defined by each size $(W_{l,d}, H_{l,d})$. This operation generates $D$ vectors that are then concatenated together into $\mathbf{p}_{i,l}$ which is the output of the proposed module.

of $S \times W \times H \times C$ slices, where $C$ is the number of image channels used to represent a slice, is given in input to a standard CNN-based pipeline (Bien et al., 2018; Tsai et al., 2020).

**Backbone.** Each slice $s_i, i \in \{0, \cdots, S-1\}$ of an MRI exam is first processed by a backbone CNN (e.g. MRNet's AlexNet) that produces feature map representations carrying semantically different information. This is achieved by considering the output feature maps of the backbone CNN's intermediate layers. More formally, we consider $\mathbf{B}_{i,l} \in \mathbb{R}^{W_l \times H_l \times C_l}$ as the output of the $l$-th layer of a backbone CNN as given by the function $f_l(s_i)$ which is inputted with $s_i$. In our setting, we consider $l \in \{1, \cdots, L\}, L = 5$, thereby exploiting 5 semantically-different CNN outputs. The execution of the backbone thus generates the set of feature maps $\{\mathbf{B}_{i,1}, \mathbf{B}_{i,2}, \mathbf{B}_{i,3}, \mathbf{B}_{i,4}, \mathbf{B}_{i,5}\}$.

**Feature Pyramid Network.** Since it has been shown that knee anomalies occupy small regions in the MRI slices (Hash, 2013; Nguyen et al., 2014; Lecouvet et al., 2018), we exploit an FPN architecture (Lin et al., 2017) which demonstrated to be particularly effective for the detection of small objects. Following Lin et al. (2017), we exploit the outputs $\{\mathbf{B}_{i,l}\}_{l=1}^{L}$ of the backbone by combining the semantically-stronger features of higher layers with the more accurately localized features of lower layers. In particular, at each level $l$, the higher-level feature maps $\mathbf{F}_{i,l+1}$ are up-sampled by bilinear interpolation and transformed by a convolutional layer and ReLU activation, to match the spatial and channel dimensions of $\mathbf{B}_{i,l}$. Then, these features are used to enhance, by element-wise sum, $\mathbf{B}_{i,l}$ which are previously transformed by a $1 \times 1$ convolution with ReLU activation (so-called lateral connection). The two steps generate the FPN features $\mathbf{F}_{i,l}$. The whole procedure is executed from the highest layer to the lowest (top-down pathway), resulting in the set of feature maps $\{\mathbf{F}_{i,l}\}_{l=1}^{L}$. At the highest level (i.e. $l = 5$) $\mathbf{F}_{i,5} = \mathbf{B}_{i,5}$ is set.

**Pyramidal Detail Pooling.** The FPN strategy enhances the quality of features for tasks that require precise spatial information, giving the freedom of designing arbitrary methods to exploit such information. We introduce PDP, a pyramidal feature pooling module capable of capturing the relevant information of the knee disorder (Hash, 2013;

Nguyen et al., 2014; Lecouvet et al., 2018). PDP analyzes the FPN representations at multiple levels of detail by focusing on increasingly smaller sub-regions localized in the slice center. The module takes as input a feature map tensor and a series of sub-regions dimensions, and produces a vectorized representation combining the information contained in each sub-region. In more details, PDP (depicted in Figure 2) is fed with $\mathbf{F}_{i,l}$ and with a list of $D \in \mathbb{N}$ sub-regions of size $(W_{l,d}, H_{l,d})$ where $d \in \{0, \cdots, D-1\}$. For each $d$, the module crops channel-wise the sub-tensor of size $(W_{l,d}, H_{l,d})$ localized at the feature maps center having coordinates $x = \lfloor \frac{W_l}{2} \rfloor, y = \lfloor \frac{H_l}{2} \rfloor$. Such sub-tensor is processed by a global pooling function $\mathcal{P}(\cdot)$ which reduces each feature map into a single value, thus obtaining a vector $\mathbf{p}_{i,l,d}$ of length $C_l$. Finally, the $D$ obtained vectors are concatenated together to obtain a single vector representation $\mathbf{p}_{i,l}$ of length $D \cdot C_l$, which corresponds to the output of the proposed module. The dimensions $(W_{l,d}, H_{l,d})$ are obtained by mapping the slice-level dimensions $(W_{0,d}, H_{0,d}), d \in \{0, \cdots, D-1\}$ following the 2D dimension reduction definitions induced by the sub-sampling operations (i.e. convolutional/pooling operations with relative kernel/stride/padding sizes) of the backbone layers. Just sub-regions resulting in $W_{l,d} > 0, H_{l,d} > 0$ are retained. If multiple slice-level sub-regions map to the same representation-level sub-region, just a single one is considered. Based on experiments, we propose a simple but effective strategy to compute sub-regions candidates in containing the knee anomaly. We generate $(W_{0,d}, H_{0,d})$ by considering

$$X_{0,d} = X \cdot (1 - \frac{d}{D}), X \in \{W, H\}, d \in \{0, \cdots, D-1\}. \tag{1}$$

In simple words, at each $d$ we consider a sub-region which focuses on an increasingly smaller part of the MRI slice, thus increasing the level of detail for that part of the slice. Defining sub-regions at the slice level simplifies the design of detailing strategies and makes the process independent from the backbone architecture. Moreover, this strategy requires the description of a single hyper-parameter $D$ to control both the number of detail levels and their size. The overall PDP strategy is designed to be robust. Indeed, if knee anomaly's features are out of the scope of an inner sub-region it can be captured by the outer sub-region of a previous detail level. At the lowest level (i.e. $d = 0$) the sub-region matches the dimensions of the backbone's feature maps, thus guaranteeing a lower-bound feature exploitation at least as good as the backbone's.

**Feature Combination and Output Prediction.** To obtain a single representation for the whole MRI exam, we follow a similar strategy to Bien et al. (2018); Tsai et al. (2020). At each backbone level $l$, a max-pooling operation is applied series-wise (therefore across slices) to $\mathbf{p}_{i,l}$. These operations result in the set of single vector representations $\{\mathbf{p}_l\}_{l=1}^L$ that summarize the information contained in the MRI sequence at multiple center-focused levels of detail. Each of these vectors is finally given to a separate fully connected layer that predicts the probability $\widehat{y}_l$ of presence/absence of a knee disorder. Based on experiments, we found the maximum of these probabilities to be the best estimate of knee anomaly presence.

**Model Learning.** For the optimization of the whole MRPyrNet pipeline, each MRI exam belonging to the training set is considered as a training sample. The probability estimates $\{\widehat{y}_l\}_{l=1}^L$ of pathology presence/absence are obtained via the procedure described in the previous sections. Each prediction is compared to the ground-truth label $y$ via the loss

function $\mathcal{L}(\widehat{y}_l, y)$ which depends on the considered backbone (details follow). The overall optimization goal is set to be $\sum_{l=1}^{L} \mathcal{L}(\widehat{y}_l, y)$.

## 3. Experimental Settings

### 3.1. Applications

We applied the MRPyrNet architecture into the pipelines of the state-of-the-art MRNet (Bien et al., 2018) and ELNet (Tsai et al., 2020) methods. For both, we kept the original pipeline configurations and hyper-parameters settings, in training and inference.

For MRNet, we considered the features $\{\mathbf{B}_{i,l}\}_{l=1}^{L}$ as the output of AlexNet's max-pool1, max-pool2, conv3, conv4, conv5 layers respectively. Each lateral connection has been implemented by a 1x1 convolutional layer with ReLU activation, and the convolutional module after the up-sampling layer has been implemented by a convolutional layer with kernel size equal to 3, and ReLU activation.[1] Each $\{\mathbf{p}_l\}_{l=1}^{L}$ is given to an independent fully connected layer which has a single output node and sigmoid activation. Values for $D$ have been configured for different MRI views after the results shown in Figure 4. The sample-distribution-weighted binary cross entropy loss has been used for $\mathcal{L}(\widehat{y}_l, y)$.

For ELNet, $\{\mathbf{B}_{i,l}\}_{l=1}^{L}$ have been considered as, respectively, the output of the first, second, third, fourth, blur-pool layers, and the output of the convolutional layer before the last blur-pooling layer. Each lateral connection has a 1x1 convolutional layer with ReLU activation, while we implemented the convolutional module after the up-sampling layer as a convolutional layer with kernel size equal to 3, ReLU activation, and the original ELNet's normalization layer.[1] Based on experiments, $D$ has been set to 6 and 7 for the ACL and meniscus tear tasks, respectively. Each output $\{\mathbf{p}_l\}_{l=1}^{L}$ is given as input to a fully connected layer with two output nodes and a softmax activation. We used the standard cross entropy loss for $\mathcal{L}(\widehat{y}_l, y)$ as done by ELNet.

### 3.2. Datasets

**MRNet Dataset.**  The MRNet dataset (Bien et al., 2018) (which we refer to as MRNet-Data) is the largest public knee MRI dataset currently available. It consists of 1370 knee MRI manually curated examinations performed at the Stanford University Medical Center in a 12-year period. Each case contains axial (proton density-weighted series), coronal (T1-weighted series), and sagittal (T2-weighted series) MRI scans obtained with GE machines. Each exam was assigned a label according to the presence/absence of ACL tear, meniscal tear, or general abnormalities that are not the before mentioned (we considered just the first two in this work). The exams were randomly split by the authors into 1130 training exams (1088 patients), 120 validation exams (111 patients), and other 120 test exams (113 patients), by making sure that each split contained at least 50 cases for each pathology. Each MRI slice is of size $256 \times 256$ pixels and their number in the sequences ranges between 17-61 (mean 31 and standard deviation 7.97).

**kneeMRI Dataset.**  The kneeMRI dataset was acquired by Štajduhar et al. (2017) at the Clinical Hospital Centre Rijeka, Croatia, from 2007 until 2014. It contains 917 sagittal

---

1. Further details are given in the Appendix B.1.

Table 1: [MRNetData Dataset] Results of baseline methodologies without and with MR-PyrNet on different diagnosis tasks. Best setups, per task and per method, are highlighted in bold.

| Diagnosis Task | Metric | Architecture | | | |
|---|---|---|---|---|---|
| | | MRNet | with MRPyrNet | ELNet | with MRPyrNet |
| ACL tear | ROC-AUC | $0.955 \pm 0.005$ | $\mathbf{0.976} \pm 0.003$ | $0.940 \pm 0.001$ | $\mathbf{0.960} \pm 0.015$ |
| | Accuracy | $0.847 \pm 0.005$ | $\mathbf{0.886} \pm 0.010$ | $0.808 \pm 0.000$ | $\mathbf{0.881} \pm 0.034$ |
| | Sensitivity | $0.722 \pm 0.000$ | $\mathbf{0.815} \pm 0.019$ | $0.648 \pm 0.019$ | $\mathbf{0.827} \pm 0.039$ |
| | Specificity | $\mathbf{0.950} \pm 0.009$ | $0.944 \pm 0.009$ | $\mathbf{0.939} \pm 0.015$ | $0.924 \pm 0.030$ |
| Meniscus tear | ROC-AUC | $0.843 \pm 0.016$ | $\mathbf{0.889} \pm 0.006$ | $0.869 \pm 0.031$ | $\mathbf{0.895} \pm 0.008$ |
| | Accuracy | $0.778 \pm 0.027$ | $\mathbf{0.808} \pm 0.008$ | $\mathbf{0.775} \pm 0.044$ | $0.761 \pm 0.042$ |
| | Sensitivity | $0.750 \pm 0.067$ | $\mathbf{0.853} \pm 0.048$ | $0.814 \pm 0.109$ | $\mathbf{0.872} \pm 0.106$ |
| | Specificity | $\mathbf{0.799} \pm 0.009$ | $0.775 \pm 0.052$ | $\mathbf{0.745} \pm 0.075$ | $0.676 \pm 0.149$ |

Table 2: [kneeMRI Dataset] Performance of baseline methodologies without and with MR-PyrNet. Best setup, per method, is highlighted in bold.

| Diagnosis Task | Metric | Architecture | | | |
|---|---|---|---|---|---|
| | | MRNet | with MRPyrNet | ELNet | with MRPyrNet |
| ACL tear | ROC-AUC | $0.902 \pm 0.005$ | $\mathbf{0.914} \pm 0.004$ | $0.873 \pm 0.020$ | $\mathbf{0.900} \pm 0.016$ |
| | Accuracy | $\mathbf{0.847} \pm 0.002$ | $0.834 \pm 0.009$ | $0.836 \pm 0.025$ | $\mathbf{0.851} \pm 0.028$ |
| | Sensitivity | $0.692 \pm 0.047$ | $\mathbf{0.806} \pm 0.008$ | $0.582 \pm 0.099$ | $\mathbf{0.679} \pm 0.089$ |
| | Specificity | $\mathbf{0.898} \pm 0.015$ | $0.843 \pm 0.013$ | $\mathbf{0.919} \pm 0.065$ | $0.908 \pm 0.065$ |

proton density-weighted exams obtained with a Siemens Avanto 1.5-T scanner. The authors, following radiologist reports, assigned each exam a label according to the level of ACL disorder: non-injured (690 exams), partially injured (172 exams), and completely ruptured (55 exams). Each MRI slice is of size $320 \times 320$ or $290 \times 300$ pixels, and the number of images in each series ranges in 21-45 (mean 31 and standard deviation 2.27). For this dataset, as done by Bien et al. (2018), we considered the classification task of discriminating between non-injured ACLs and injured ACLs.

### 3.3. Performance Evaluation and Measures

The performance evaluation on MRNetData dataset has been executed on the validation set (since the test set is sequestered) after optimizing the model on the training set. For the kneeMRI dataset, performance was assessed through a 5-fold cross-validation procedure by considering, in each fold, 80% of the exams for training and the remaining 20% for validation. As quantitative measures, we used the area-under-the-curve of the receiver operating characteristics (ROC-AUC), the accuracy, and the sensitivity and specificity obtained after thresholding the probability predictions at 0.5. Each experiment has been run three times with the different random seeds (the same three values for each experiment). We report on the mean and standard deviation for each metric.

### 3.4. Implementation Details

Code[2] was implemented in Python with the PyTorch (Steiner et al., 2019) and scikit-learn (Pedregosa et al., 2011) machine learning frameworks. MRNet and ELNet have been implemented using the code published by the authors and by following the details of the respective papers. A machine with an Intel Xeon E5-2690 v4 @ 2.60GHz CPU, 320 GB of RAM, and an NVIDIA TITAN V GPU has been employed to run the experiments.

---

2. https://git.io/JtMPH

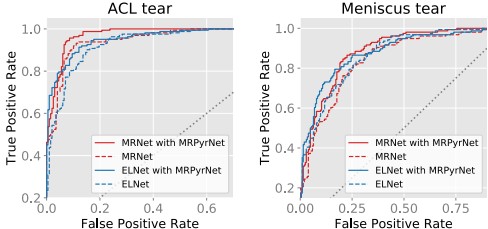
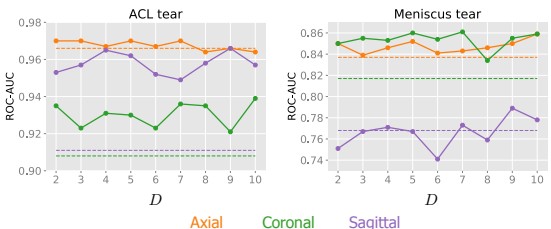

Figure 3: ROC curves showing the performance of the baseline methods with and without MRPyrNet.

Figure 4: PDP's performance under different $D$ settings for each of the MRI views. Dashed lines show the original MRNet's results.

Table 3: ROC-AUC performance of MRPyrNet in comparison with baselines that extract information of the disorder's anatomy from different sources. Best method, per task, is highlighted in bold.

| Baseline | Diagnosis Task | |
|---|---|---|
| | ACL tear | Meniscus tear |
| slice-level pyramid | $0.969 \pm 0.002$ | $0.865 \pm 0.005$ |
| PDP on last layer | $0.971 \pm 0.007$ | $0.859 \pm 0.009$ |
| MRPyrNet | $\textbf{0.972} \pm 0.006$ | $\textbf{0.879} \pm 0.010$ |

Table 4: ROC-AUC performance achieved by MRPyrNet by removing its key modules. Best setting result, per task, is highlighted in bold.

| Components | | Diagnosis Task | |
|---|---|---|---|
| FPN | PDP | ACL tear | Meniscus tear |
| | | $0.943 \pm 0.006$ | $0.828 \pm 0.013$ |
| ✓ | | $0.960 \pm 0.005$ | $0.859 \pm 0.002$ |
| | ✓ | $0.971 \pm 0.007$ | $0.855 \pm 0.009$ |
| ✓ | ✓ | $\textbf{0.972} \pm 0.006$ | $\textbf{0.879} \pm 0.010$ |

## 4. Results

**General Remarks.**  Table 1 and Figure 3 report on the MRNetData dataset performance using MRPyrNet on the top of MRNet and ELNet pipelines.[3]  Our proposed strategy improves the ACL tear and meniscus tear detection of the baselines, showing that focusing on specific features of the disorder's anatomy is of paramount importance to improve the diagnosis capabilities. Specifically, for the ACL tear, the ROC-AUC improvement of MRPyrNet over the MRNet baseline is 2.2%, and 2.1% over the ELNet baseline. For meniscal tear, using MRPyrNet improves the score by 5.5% and 2.9% respectively. More importantly, the proposed solution enhances the sensitivity of the methods, thus their ability to detect disorders when these are present. Indeed, we observe an improvement of 12.9% and 28% over MRNet and ELNet respectively in the ACL tear detection, and of 13.9% and 7.1% in the diagnosis of the meniscus tear. The specificity decreases by a small margin in general, but it is consistent with the aim of MRPyrNet which is to extract relevant information of the disorder's anatomy when this exists. Similar conclusions can be reached for the results presented in Table 2, which show that the capabilities introduced by MRPyrNet generalize well to similar tasks but with different data distributions.

**Analysis.**  In this section, we provide a study on the sensibility of MRPyrNet to different configuration settings. The results presented in this paper have been obtained on the MRNetData dataset using the MRPyrNet-enhanced MRNet with $D = 5$.

Table 3 reports the performance of our proposed strategy against MRNet which is executed on multiple sub-regions extracted at slice-level, and against MRNet with PDP

---

3. See the Appendix C for statistical tests.

Table 5: ROC-AUC scores of MRPyrNet with different output selection strategies. Best strategy, per task, is highlighted in bold.

| Exit Combination | Diagnosis Task | |
| --- | --- | --- |
| | ACL tear | Meniscus tear |
| fully connected layer | $0.971 \pm 0.010$ | $0.851 \pm 0.012$ |
| avg | $0.970 \pm 0.003$ | $0.872 \pm 0.011$ |
| max | $\mathbf{0.972} \pm 0.006$ | $\mathbf{0.879} \pm 0.010$ |

Table 6: ROC-AUC results of MRPyrNet with different pooling functions $\mathcal{P}(\cdot)$ for PDP. Best results, per task, are highlighted in bold.

| $\mathcal{P}(\cdot)$ | Diagnosis Task | |
| --- | --- | --- |
| | ACL tear | Meniscus tear |
| max | $0.972 \pm 0.005$ | $0.849 \pm 0.011$ |
| avg | $\mathbf{0.972} \pm 0.006$ | $\mathbf{0.879} \pm 0.010$ |

applied to its last layer's feature maps. [4] MRPyrNet's performance is higher, but the two results confirm that extracting particularly localized features of the knee disorder is relevant, as both strategies improve MRNet.

Figure 4 shows the performance of MRPyrNet on different MRI views and for different values of $D$. For the ACL tear, our solution improves MRNet for all $d$ mainly on the coronal and sagittal views. For the detection of meniscal tears, results on axial and coronal views are always improved. The performance on the sagittal axis increase by using larger $D$ values. In general, these results suggest that, to achieve better accuracy, the $D$ values must be selected for the sub-regions sizes they generate rather than for the increased number.

In Table 4 we present an ablation study over MRPyrNet's components. As a baseline (first row), we consider MRPyrNet whose $\{\mathbf{B}_{i,l}\}_{l=1}^{L}$ are reduced by a global average pooling before performing the series-wise max pooling. Employing just the FPN architecture improves the baseline, showing that it enables the exploitation of small-appearing features. Introducing just the PDP module also improves the baseline performance, demonstrating that the capabilities of MRPyrNet are not just due to the increased expressive power given by the FPN's weights. The strongest strategy is achieved when both the two components are exploited. Table 5 shows that taking the max of the output probabilities given by MRPyrNet results better than computing their average or combining them via a fully connected layer. Table 6 reveals that implementing $\mathcal{P}(\cdot)$ as an avg rather than a max operation is a better choice for the detection of meniscal tear, whereas there is no particular difference for the detection of ACL tears. This is consistent with the MRI appearance of such tears. Indeed, ACL tears cause higher signal (Kam et al., 2010) that can captured also with max operations, while meniscus tears present more balanced signals (Nguyen et al., 2014) that can be better summarized with avg operations.

## 5. Conclusions

In this paper, we present MRPyrNet, a CNN architecture to extract more relevant features of knee injuries imaged with MRI. Different works in the literature showed that particular disorders are localized in small-sized knee regions that appear around the center of MRI scans. Through an FPN and a PDP module, MRPyrNet is capable of capturing the information in such areas more effectively. The proposed solution is general and, in this paper, it has been applied to the state-of-the-art diagnosis pipelines of MRNet and ELNet. Extensive experiments performed on two publicly available datasets demonstrate the effectiveness of the strategy in providing better diagnoses of ACL tear and meniscus tear.

---

4. For the details on the compared baselines please see the Appendix D.

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

## Appendix A. Related Work

**Deep Learning in the Knee.** Deep learning methodologies have been exploited for different medical tasks on the knee. Classification-based and segmentation-based CNNs were used to assess lesions of the knee cartilage on MRI images (Liu et al., 2018). On the same knee structure but imaged with ultrasound, U-Net (Ronneberger et al., 2015) based solutions have been used for knee femoral cartilage segmentation (Antico et al., 2020) and tracking (Dunnhofer et al., 2020) during minimally invasive surgeries. 3D CNNs were proposed to increase the resolution of knee MRI images (Chaudhari et al., 2018), while deep siamese CNNs were shown to score effectively the severity of knee osteoarthritis (Tiulpin et al., 2018). Liu et al. (2019a) employed a cascade of CNNs to diagnose ACL tears, while Bien et al. (2018), Tsai et al. (2020), and Azcona et al. (2020) showed that single CNNs for image classification are general enough to detect disparate abnormalities of the knee including ACL and meniscal tears. Our proposed MRPyrNet solution addresses the same problem tackled by the latter methods, but introduces a new CNN architecture that takes into account the anatomy of knee disorders to better capture its appearance features.

**Feature Pyramids.** Feature pyramids have been extensively used in computer vision to tackle different problems related to object detection. In the setting of natural images, feature pyramid networks were exploited to learn more efficient image representations (Lin et al., 2017; Kirillov et al., 2019). Towards more application-specific solutions, feature pyramids have been used for shadow detection (Zhu et al., 2018), pavement crack detection (Yang et al., 2020), land segmentation (Seferbekov et al., 2018), person re-identification (Martinel et al., 2019), person pose estimation (Chen et al., 2018), and multi-object tracking (Lee and Kim, 2019). In the context of medical image analysis, feature pyramids have been studied for the detection of pulmonary pathologies (Xiao et al., 2019) and pulmonary nodules (Liu et al., 2019b), or for the diagnosis of cervical cancer (Xu et al., 2015) and lung carcinoma (Hao et al., 2020). Currently, no method is present to tackle the problem of knee pathology detection through the adoption of feature pyramids.

## Appendix B. Details on the Experiments

### B.1. Architectural Details of the FPN

Table 7 presents the architectural details employed in the proposed FPN for each of the experimented baseline pipeline.

### B.2. Details on the Application of MRPyrNet

In this section, we give more details on how we applied the proposed MRPyrNet strategy to the baselines MRNet (Bien et al., 2018) and ELNet (Tsai et al., 2020). Following the details of MRNet's original paper for the MRNetData dataset, we first trained a single-view model (i.e. for each of the axial, coronal, and sagittal views) with MRPyrNet applied for each diagnostic task. Then, the probability predictions of the three models have been combined with a logistic regressor to obtain a single probability estimate of presence/absence of the knee disorder for each of the training exams. The optimization of the models has been performed using the original settings. The Adam optimizer (Kingma and Ba, 2014) has

Table 7: Architectural details of the FPN architecture employed by MRPyrNet for different backbones. For each baseline pipeline (MRNet, ELNet), we report: the size $W_l \times H_l \times C_l$ of the feature maps at each backbone level $l$; the layer configuration of the lateral connection (kernel size and number of channels of the convolutional layers, and activation function); the layer configuration of the up-sampling module (kernel size and number of channels of the convolutional layers, activation functions, and eventual normalization layers - "bil interp" stands for the bilinear interpolation operation).

| Backbone | Module | Backbone Level | | | | |
|---|---|---|---|---|---|---|
| | | $l = 1$ | $l = 2$ | $l = 3$ | $l = 4$ | $l = 5$ |
| MRNet | Output Size | $31 \times 31 \times 64$ | $15 \times 15 \times 192$ | $15 \times 15 \times 384$ | $15 \times 15 \times 256$ | $7 \times 7 \times 256$ |
| | Lateral Connection | Conv 1x1 64 - 64
ReLU | Conv 1x1 192 - 192
ReLU | Conv 1x1 384 - 384
ReLU | Conv 1x1 256 - 256
ReLU | -
- |
| | Up-sampling | -
-
-
-
- | bil interp
Conv 1x1 192 - 64
ReLU
Conv 3x3 64 - 64
ReLU | bil interp
Conv 1x1 384 - 192
ReLU
Conv 3x3 192 - 192
ReLU | bil interp
Conv 1x1 256 - 384
ReLU
Conv 3x3 384 - 384
ReLU | bil interp
Conv 1x1 256 - 256
ReLU
Conv 3x3 256 - 256
ReLU |
| ELNet | Output Size | $62 \times 62 \times 16$ | $29 \times 29 \times 32$ | $13 \times 13 \times 64$ | $5 \times 5 \times 64$ | $5 \times 5 \times 64$ |
| | Lateral Connection | Conv 1x1 16 - 16
ReLU | Conv 1x1 32 - 32
ReLU | Conv 1x1 64 - 64
ReLU | Conv 1x1 64 - 64
ReLU | -
- |
| | Up-sampling | -
-
-
-
- | bil interp
Conv 1x1 32 - 16
ReLU
Conv 3x3 16 - 16
Norm
ReLU | bil interp
Conv 1x1 64 - 32
ReLU
Conv 3x3 32 - 32
Norm
ReLU | bil interp
Conv 1x1 64 - 64
ReLU
Conv 3x3 64 - 64
Norm
ReLU | bil interp
Conv 1x1 64 - 64
ReLU
Conv 3x3 64 - 64
Norm
ReLU |

been employed with a learning rate of $10^{-5}$ that was decreased by 0.3 in the presence of plateaux of the validation performance while executing 50 epochs. The original data augmentation strategy that applies random shift up to 25 pixels, random rotation up to 25 degrees, and random horizontal flip of an MRI exam, has been implemented. Similar optimization settings have been employed for the experiments on the KneeMRI dataset. In this case, we trained just an instance of MRNet with MRPyrNet since this dataset offers just sagittal MRI scans. A weight decay with $10^{-2}$ factor was added to the optimization objective. All the MRI slices have been resized to $256 \times 256$ pixels. Data augmentations included random scale with a random factor in $[0.9, 1.1]$, random shift up to 40 pixels, and random rotation up to 25 degrees.

For the application of MRPyrNet to ELNet's backbone, we trained an instance on the axial view for the ACL tear, and another on the coronal view for the meniscus tear tasks based on the MRNetData dataset. The optimization of the models has been performed using the original settings. The Adam optimizer (Kingma and Ba, 2014) has been employed with a learning rate of $2^{-5}$ for the ACL tear task, and of $1.5^{-5}$ for the meniscal tear task. The training was run for 200 epochs. Layer normalization (Ba et al., 2016) has been set for the ACL tear, and contrast normalization (Ulyanov et al., 2016) for the detection of meniscus tear, both with the factor value $K$ equal to 4. The original data augmentation strategy has been applied, including random shift up to 25 pixels, random scaling with a random factor chosen in the range $[0.9, 1.1]$, random rotation up to 10 degrees, random horizontal flip, random rotation by a multiple of 90 degrees. For the experiments on the KneeMRI dataset, the SGD optimizer with a momentum of 0.9 has been employed with a learning rate of $5^{-5}$. Dropout layers have been added with a 0.5 factor as for the original ELNet. All the MRI slices have been resized to $256 \times 256$ pixels. The implemented data augmentation strategy included random scale with a random factor in $[0.9, 1.1]$, random shift up to 25 pixels, and random rotation up to 10 degrees.

## Appendix C. Statistical Tests

We performed a McNemar's test (McNemar, 1947) to assess the significance of the proposed methodology's performance. The p-value obtained from the sensitivity comparison between MRNet and MRNet with MRPyrNet results in 0.0003 and 0.0005 for the ACL tear and meniscus tear respectively. The p-value for the sensitivity comparison between ELNet and ELNet with MRPyrNet results in 0.0003 and 0.016 for the ACL tear and meniscus tear respectively.

## Appendix D. Details on compared Baselines that use Disorder's Information.

We compare our MRPyrNet to other baseline strategies that aim to extract information of the knee disorder anatomy. The first baseline (first row of Table 3) generates a list of sub-region sizes $(W_d, H_d), D = 5$, at slice-level. These are used to crop each slice $s_i$ at its center coordinates. Each cropped slice is resized to $256 \times 256$ pixels and given to MRNet's AlexNet which produces $D$ feature vectors (one for each cropped slice). These vectors are concatenated together before the application of the series-wise max-pooling operation. After that, a fully connected layer predicts the probability of disorder presence/absence. The second baseline (second row of Table 3) just applies the proposed PDP to the feature-maps $\mathbf{B}_{i,5}$ given by MRNet's AlexNet before the series-wise max-pooling.

## Appendix E. Additional Results

Since MRPyrNet exploits features localized in particular areas of the MRI slices, we performed an experiment to assess its robustness to vertical and horizontal shifts of the slices, since these events could change the spatial localization of such features and consequently influence MRPyrNet's capabilities. Specifically, we applied random vertical and horizontal translations up to the 20% of the slice sizes (around 50 pixels) on the MRI scans belonging to the MRNetData validation set. In such a setting, MRPyrNet applied to the baseline MRNet achieves a ROC-AUC and sensitivity of $0.967 \pm 0.009$ and $0.765 \pm 0.021$ for the ACL tear detection, and a ROC-AUC and sensitivity of $0.879 \pm 0.005$ and $0.840 \pm 0.044$ for the meniscal tear. These results are a little lower than the ones presented in Table 1, but remain higher than the ones of the baseline MRNet. We also performed the same experiment over the original MRNet. Such a baseline achieves a ROC-AUC and sensitivity of $0.951 \pm 0.001$ and $0.685 \pm 0.032$ for the ACL tear detection, and a ROC-AUC and sensitivity of $0.839 \pm 0.015$ and $0.718 \pm 0.044$ for the meniscal tear detection. Even these results are lower than the original baseline whose results are also available in Table 1. Overall, these outcomes show that part of the error committed by MRPyrNet is inherited from the baseline architecture. Nevertheless, given the limited performance drop, we can state that MRPyrNet is rather robust to the spatial location change of abnormalities.

## Appendix F. Limitations

Tables 1 and 2 show that the application of MRPyrNet reduces the specificity in general. We think that this is due to MRPyrNet's architecture which sets an inductive bias on better

representing features related to abnormalities. For example, the employment of the FPN sets a bias over the exploitation of small appearing features. In this sense, MRPyrNet is designed to exploit the information about the abnormalities only when these are actually present. Future work will be devoted to reduce the gap in the specificity. Moreover, we would like to remark that our proposed PDP strategy should be considered as a generic baseline to implement a prior over the knee disorders' anatomy. Indeed, based on the observation that abnormalities appear in particular areas of the MRI slices, we demonstrated that our PDP module captures relevant information that leads to the improvement of the diagnoses. We think that more sophisticated strategies exploiting additional cues on the knee's anatomy could additionally enhance the results presented in this paper.

