# OpenReview forum: "Improving MRI-based Knee Disorder Diagnosis with Pyramidal Feature Details"
_MIDL.io/2021/Conference — MIDL 2021_

### Official Review · AnonReviewer3 · 2021-03-06

**Confidence:** 4
**Preliminary Rating:** 3
**Recommendation:** Poster
**Final Rating:** 4

**Summary:**

The authors presents an approach for detecting ACL tear and meniscal
tear in knee MRI scans. The approach is based on MRNet and uses the problem
insight, that most of the relevant information lies towards the center crop region of the
MRI scan. To leverage that, they propose using a Feature Pyramid Network, because
of its hierarchical structure and furthermore introduce the Pyramidal Detail Pooling
(PDP) operation. They evaluate their method on two publicly available datasets.

**Strengths:**

1. The paper is clearly written and formatted.
2. The approach is simple and well-motivated.
3. The authors evaluate against two recent papers on this task, observing a significant increase in performance
4. A thorough evaluation of different parts of the approach is made.
5. Code will be made available.

**Weaknesses:**

The presented approach is an application of known methods to a new problem, only introducing marginally new ideas. In an method driven conference, such lack of methodological novelty would have been an issue. But since MiDL has a strong application focus, I am inclined towards seeing this paper be presented. However, I would only like to see such a paper be presented in the application track of MiDL.

**Deanonymize Review:**

no

**Detailed Comments:**

1. The presented approach is simple and well-motivated and uses tested architectures and techniques. By evaluating against two prior works, the authors show, that their approach achieves significantly higher performance metrics.

2. Section 4 provides a detailed analysis of different subparts of MRPyrNet, showing the importance of the PDP pooling.

3. The paragraph on the selection of subregions in section 2 on the Pyramidal Detail Pooling is confusing. Maybe a revision of these sentences would make it easier to understand.

**Final Rating Justification:**

I thank the authors for their answers. My questions have been answered, so I will change my score to ‘Strong Accept’.'

**Justification Of The Preliminary Rating:**

Although the paper lacks in general novelty, the authors present a good working approach with a wide range of tests and ablation studies. Since MiDL has a strong application focus, I am inclined towards seeing this paper be presented. That's why I am leaning towards acceptance. However, the authors must answer my questions adequately in the rebuttal.

**Paper Type:**

validation/application paper

**Questions To Address In The Rebuttal:**

1. Do you have an explanation for why MRPyrNet performs worse on the Meniscus Tear benchmark compared to ELNet but better on ACL Tear?
2. In section 3.3 it says that ‘multiple runs’ where used to compute the std of the metrics. How many runs where used for the evaluation?
3. A surprising result is that different numbers D of pooled regions (Figure 4) does not seem to have a big effect on ACL tear detection, while not having it (Figure 4 and Table 4) seems to have a big effect. Can you elaborate on that?
4. Have the authors tried different approaches for selecting the region sizes in their PDP approach?
5. Have the authors tried (following their motivation for using the PDP), dropping coarser image regions, basing the classification entirely on the center crops?


**Special Issue:**

no

---

> ### Author Response · Authors · 2021-03-17
> **Replies to AnonReviewer3**
>
> We thank the reviewer for his/her insightful comments and suggestions.
>
> > Do you have an explanation for why MRPyrNet performs worse on the Meniscus Tear benchmark compared to ELNet but better on ACL Tear?
>
> We think that the higher performance achieved on the ACL tear task is due to the fact that such a tear is better distinguishable from its context because of its appearance with higher pixel intensities (Kam et al., 2010). This leads to better discriminative backbone features which in turn are better exploited by MRPyrNet. On the other hand, since meniscal tears appear with a more balanced distribution of frequencies (Nguyen et al., 2014), it results more difficult to separate an abnormality from the surrounding context. This is confirmed by the ROC-AUC scores for the meniscal tear task which are generally much lower than the ones achieved on the ACL tear. In this context, we think that the backbone features produced by ELNet are even less discriminative, as it employs Blur pooling as a downsampling operation which was discussed to cause over filtering (Zhang, 2019). Due to its inductive bias (which we discussed with AnonReviewer1) that favors the increase of the sensitivity, MRPyrNet is able to capture the most relevant signal when abnormalities are present, but it is confused on ambiguous negative cases that might no present a significant difference from the features of positive samples.
>
> > In section 3.3 it says that ‘multiple runs’ where used to compute the std of the metrics. How many runs where used for the evaluation?
>
> For each experiment discussed in our paper, 3 runs with different random seeds were performed, and the same 3 random seed values were used for all the experiments. This detail has been added in the revised version of the paper.
>
> > A surprising result is that different numbers D of pooled regions (Figure 4) does not seem to have a big effect on ACL tear detection, while not having it (Figure 4 and Table 4) seems to have a big effect. Can you elaborate on that?
>
> This behavior can be attributed to the fact that the global average pooling of the original MRNet applied to the highest layer compresses too much information. Indeed, just the application of PDP to such a layer (as presented in row 2 of Table 3) is able to improve the detection performance of the baseline. Therefore, even having a PDP module with a low number of $D$ (e.g. = 2) improves the performance because it is able to retain more information regarding the slices.
>
> > Have the authors tried different approaches for selecting the region sizes in their PDP approach?
>
> We tried to generate the RoIs with an exponential formulation, substituting Eq. (1) with $X_{0,d} = \frac{X}{2^{d}}, X \in \\{W,H\\},d \in \\{ 0, \cdots, d-1 \\}$.
> However, the ROC-AUC scores achieved were $0.968 \pm 0.008$ and $0.875 \pm 0.008$ (with $D = 5$) for the ACL tear and meniscus tear respectively. These results, which are lower than the ones reported in our paper, made us favor the proposed linear strategy as a first step towards a solution. In future work, our plan is to study alternative strategies to generate RoIs, as well as other RoI shapes (e.g. circles).
>
> > Have the authors tried (following their motivation for using the PDP), dropping coarser image regions, basing the classification entirely on the center crops?
>
> We did not try such an approach because we think it would harm the robustness of the proposed solution. Indeed, the PDP module extracts multiple RoIs that increasingly focus on the image center. Doing this way, if a lesion is out of view of an inner RoI it can be still captured by the outer RoI of a previous detail level, which analyzes a larger slice area. At the lowest detail level, the RoI matches the size of the backbone's feature maps. This setting allows to look for more detailed features while guaranteeing that the abnormality is always captured by at least one detail level, ultimately making the PDP module more robust.

---

### Official Review · AnonReviewer4 · 2021-03-07

**Confidence:** 4
**Preliminary Rating:** 3
**Recommendation:** Oral
**Final Rating:** 3

**Summary:**

The paper describes a Convolutional Neural Network (CNN) architecture for the diagnosis of knee injury from MRI scans. The CNN architecture is composed of a Feature Pyramid Network (FPN) with Pyramidal Detail Pooling (PDP), which allows the network to focus on sub-regions of varying size centred on the image. The presented results demonstrate improved performance as a result of the proposed techniques.

**Strengths:**

1. The paper is well structured and presented with sufficient details of methods and results.
2. The proposed architecture and techniques bear similarity with multiple existing architectures (e.g., encoder-decoder architecture with skip connection similar to UNet), but have some novel aspects.
3.  The results are compared with proper baseline methods and discussed properly.

**Weaknesses:**

1. The proposed technique termed FPN is applied on a prior work’s (MRNet) architecture as the backbone. Based on the description provided in the paper, it seems that the application of proposed techniques to any backbone architecture will increase the depth and parameters of the neural network considerably. As mentioned by the authors, the prior work (MRNet) uses the famous AlexNet architecture, which is not as deep as other succeeding and better performing classification architectures e.g., VGGNet, ResNet, DenseNet. This raises an obvious question: Is the increase in performance compared to backbone architecture merely a result of increased network capacity?
Therefore, it is important to apply the proposed techniques with more modern architectures as backbone architectures and see if the improvement in performance by application of the proposed technique is consistent across architectures.
2. The authors mention that the relevant features are localized near the centre of MRI scans. This assumption as such assumes that all the knee MRI scans are acquired centred on the knee. This overlooks the possibility of translation during MRI acquisition due to known or unknown reasons.

**Deanonymize Review:**

no

**Detailed Comments:**

The proposed method seems to have lower specificity than the baseline methods in both datasets. Could it be an indication of a wrong operating point chosen? The choice of operating point should be decided and discussed based on the underlying clinical requirements (some clinical settings require higher sensitivity while others require higher specificity).

**Final Rating Justification:**

The authors have addressed all the concerns raised by me to my satisfaction. However, I kept the rating the same as previous because the level of methodological development presented in the paper is not strikingly novel.

**Justification Of The Preliminary Rating:**

The paper presents a Feature Pyramid Network (FPN) with Pyramidal Detail Pooling (PDP) applied on any backbone CNN architecture. The proposed method has some novel aspects: using skip connections to combine top-down and bottom-up features, pyramid detail pooling on features maps containing different level of semantic information. The presented results are also convincing. However, the backbone architecture used is not so deep compared to many modern classification architectures. It might be possible that the observed performance gain is a result of increased network capacity by the addition of more depth to the backbone architecture rather than the proposed technique itself. The paper lacks additional experiments to provide insights in this direction.

**Paper Type:**

methodological development

**Special Issue:**

no

---

> ### Author Response · Authors · 2021-03-17
> **Replies to AnonReviewer4**
>
> We thank the reviewer for his/her wise comments.
>
> > The proposed technique termed FPN is applied on a prior work’s (MRNet) architecture as the backbone. Based on the description provided in the paper, it seems that the application of proposed techniques to any backbone architecture will increase the depth and parameters of the neural network considerably. As mentioned by the authors, the prior work (MRNet) uses the famous AlexNet architecture, which is not as deep as other succeeding and better performing classification architectures e.g., VGGNet, ResNet, DenseNet. This raises an obvious question: Is the increase in performance compared to backbone architecture merely a result of increased network capacity? Therefore, it is important to apply the proposed techniques with more modern architectures as backbone architectures and see if the improvement in performance by application of the proposed technique is consistent across architectures.
>
> We investigated the employment of deeper networks at the beginning of our research project on this topic. We found deeper networks (like ResNet50) to work comparably to AlexNet on the MRNetData dataset, but to dramatically suffer from overfitting in the experiments over the kneeMRI dataset. Thus we favoured less deep networks for the better generalization across domains. This is consistent with the outcomes achieved by the state-of-the-art solutions (Bien et al. 2018, Tsai et al. 2020, Azcona et al. 2020) which all employ smaller networks for these diagnostic tasks.
> Moreover, row 3 of Table 4 shows that the performance of the baseline backbone can be improved just with the PDP module, without the addition of learnable parameters. Such an outcome shows that the increased expressive power is not enough for better performance, but specific information about the diagnostic domain has to be exploited.
>
> > The authors mention that the relevant features are localized near the centre of MRI scans. This assumption as such assumes that all the knee MRI scans are acquired centred on the knee. This overlooks the possibility of translation during MRI acquisition due to known or unknown reasons.
>
> As posted in the reply to reviewer AnonReviewer1, we performed an experiment to assess the robustness of our solution to vertical and horizontal shifts of the MRI slices. Specifically, we applied random vertical and horizontal translations up to the 20% of the slice sizes on the test MRI scans. In such a setting, MRPyrNet applied to the baseline MRNet achieves a ROC-AUC and sensitivity of $0.967 \pm 0.009$ and $0.765 \pm 0.021$ for the ACL tear detection, and a ROC-AUC and sensitivity of $0.879 \pm 0.005$ and $0.840 \pm 0.044$ for the detection of meniscal tears. These results are a little lower than the ones presented in Table 1, but remain higher than the ones of the baseline MRNet. We also performed the same experiment over the original MRNet. Such a baseline achieves a ROC-AUC and sensitivity of $0.951 \pm 0.001$ and $0.685 \pm 0.032$ for the ACL tear detection, and a ROC-AUC and sensitivity of $0.839 \pm 0.015$ and $0.718 \pm 0.044$ for the meniscal tear detection. Even these results are lower than the original baseline whose results are also available in Table 1. Overall, these outcomes show that part of the error committed by MRPyrNet is inherited from the baseline architecture. Nevertheless, given the limited performance drop, we can state that MRPyrNet is rather robust to the visual translation of abnormalities.
>
> > The proposed method seems to have lower specificity than the baseline methods in both datasets. Could it be an indication of a wrong operating point chosen? The choice of operating point should be decided and discussed based on the underlying clinical requirements (some clinical settings require higher sensitivity while others require higher specificity).
>
> We agree with the reviewer on the fact that the operating point should be chosen according to clinical requirements. However, we selected the threshold 0.5 to have a fair comparison with MRNet and ELNet, which employed such a value to compute sensitivity and specificity. Comparing to those methods in the described setting allows us to state that MRPyrNet is better suited for clinical applications that require higher sensitivity. Nevertheless, Figure 3, by reporting ROC curves, shows ROC scores obtained under different thresholds. Hence, it can be used to get insights on the operating point selection (at least for the sensitivity).

---

### Official Review · ~Simeon_Emilov_Spasov1 · 2021-03-08

**Confidence:** 4
**Preliminary Rating:** 3
**Recommendation:** Poster
**Final Rating:** 3

**Summary:**

The authors propose an augmentation to existing deep learning frameworks for knee disorder prediction. They incorporate the prior knowledge that anomalies are localized in small central locations of knee MRI scans. The authors exploit this knowledge by using Feature Pyramid Networks (FPNs) and Pyramid Detail Pooling (PDP). The FPN allows the proposed framework to combine extracted activations from a backbone CNN architecture from layers at different depths. Then, the PDP crops central sub-regions of different spatial scales from the FPN activations. When baseline CNNs are augmented with the proposed framework, results are consistently better.

**Strengths:**

1.	The paper is trying to exploit a structural prior in MRI scans specific to the anatomy of knee disorders. I find this idea *interesting in itself* although I wonder if the implementation per se (assumption that key features are centered on MRI scans) can be consistently relied upon in clinical practice.
2.	Robust testing procedure by 5-fold cross-validation. Repeatable results with statistical significance.
3.	Both table 1 and 2 show consistently better results with the MRPyrNet modifications on the baseline networks (especially ROC-AUC metric).  Results also corroborated by statistical tests.
4.	The paper also provides thorough ablation studies. E.g. table 4 motivates each component in the system. Same for tables 5 and 6 – they motivate the use of max probability for the outputs and max pooling.


**Weaknesses:**

1.	Subsection “Feature Combination and Output Prediction” not very clear. Is the max-pooling of p_(i,l) performed across slices to form a single p_l vector?
2.	Table 4 suggests the proposed method consistently underperforms in specificity although other metrics are higher compared to baselines. The authors apply a fully connected layer at each p_l where l = 1…L to obtain y_hat_l. Then, they state they use the highest y_hat_l as a predictor. Do the authors think using the max y_hat_l could be the reason for the underperforming specificity, i.e. high sensitivity is achieved because of overestimating the probability of disease (although max probability gives highest accuracy as suggested by table 5)?
3.	It is not clear to me if the method can handle vertical/horizontal translation? It seems overly reliant on the fact that certain knee features will be localized in the center of the MRI image. Is this standard medical practice in image acquisition for knee disorder diagnosis? Would MRI centering require additional effort by clinicians?


**Deanonymize Review:**

yes

**Final Rating Justification:**

I would like to thank the authors for performing the additional experiment of translating the images and producing results against the baselines. I believe they make sense, given the key assumption of the paper, and are a good addition to the paper overall. I also think the idea of incorporating structural image priors about the location of telltale signs of disorders to be interesting although in this paper only a specific case is presented where the focus is simply the center of the image. Overall, I recommend acceptance of this manuscript.

**Justification Of The Preliminary Rating:**

In my opinion, the strengths of the paper outweigh the negatives (heavy reliance on knee features centered in MRI image and slight lack of clarity in section “Feature Combination and Output Prediction”). The method is well motivated, results against baselines are good and the ablation studies are needed and convincing. I would like to ask the authors to address the assumption that knee features are practically always centered in MRI images for knee disorder diagnosis.

**Paper Type:**

methodological development

**Questions To Address In The Rebuttal:**

Could the authors please comment on the assumption that certain knee features are *always* centered in MRI images for knee disorder diagnosis?

**Special Issue:**

no

---

> ### Author Response · Authors · 2021-03-17
> **Replies to AnonReviewer1**
>
> The authors thank the reviewer for his/her precious comments.
>
> > Subsection “Feature Combination and Output Prediction” not very clear. Is the max-pooling of p_(i,l) performed across slices to form a single p_l vector?
>
> Yes, the max-pooling is performed across slices to obtain a single representation for the whole MRI volume (as suggested by MRNet and ELNet) at each backbone level $l$. In our revised paper, the text has been improved to better explain this operation.
>
> > Table 4 suggests the proposed method consistently underperforms in specificity although other metrics are higher compared to baselines. The authors apply a fully connected layer at each p_l where l = 1…L to obtain y_hat_l. Then, they state they use the highest y_hat_l as a predictor. Do the authors think using the max y_hat_l could be the reason for the underperforming specificity, i.e. high sensitivity is achieved because of overestimating the probability of disease (although max probability gives highest accuracy as suggested by table 5)?
>
> We think the lower performance in specificity is due to the MRPyrNet's architecture which sets an inductive bias to better represent features related to abnormalities. For example, the employment of the FPN sets a bias over the exploitation of small appearing features. In this sense, MRPyrNet is designed to exploit the information about the abnormalities, and this can be done effectively only when abnormalities are present.
>
> > It is not clear to me if the method can handle vertical/horizontal translation? It seems overly reliant on the fact that certain knee features will be localized in the center of the MRI image. Is this standard medical practice in image acquisition for knee disorder diagnosis? Would MRI centering require additional effort by clinicians?
>
> To answer this question, we performed an experiment to assess the robustness of our solution to vertical and horizontal shifts of the MRI slices. Specifically, we applied random vertical and horizontal translations up to the 20% of the slice sizes on the test MRI scans. In such a setting, MRPyrNet applied to the baseline MRNet achieves a ROC-AUC and sensitivity of $0.967 \pm 0.009$ and $0.765 \pm 0.021$ for the ACL tear detection, and a ROC-AUC and sensitivity of $0.879 \pm 0.005$ and $0.840 \pm 0.044$ for the meniscal tear. These results are a little lower than the ones presented in Table 1, but remain higher than the ones of the baseline MRNet. We also performed the same experiment over the original MRNet. Such a baseline achieves a ROC-AUC and sensitivity of $0.951 \pm 0.001$ and $0.685 \pm 0.032$ for the ACL tear detection, and a ROC-AUC and sensitivity of $0.839 \pm 0.015$ and $0.718 \pm 0.044$ for the meniscal tear detection. Even these results are lower than the original baseline whose results are also available in Table 1. Overall, these outcomes show that part of the error committed by MRPyrNet is inherited from the baseline architecture. Nevertheless, given the limited performance drop, we can state that MRPyrNet is, to some extent, robust to the spatial location change of abnormalities.
>
> As correctly pointed out by the reviewer, we would like to remark that our approach is a proof of concept of the idea of extracting more relevant information using priors on the diagnosis task. Our PDP module is designed by the observation that abnormalities are often (not always) located in particular sub-regions of the slices, and the results we achieved confirm our thesis. However, our approach is general enough that it enables the development of different strategies for the extraction of relevant information regarding the abnormalities.
>
> > Could the authors please comment on the assumption that certain knee features are always centered in MRI images for knee disorder diagnosis?
>
> Please see our previous reply.

---

### Official Review · AnonReviewer2 · 2021-03-09

**Confidence:** 2
**Preliminary Rating:** 2
**Recommendation:** Poster
**Final Rating:** 3

**Summary:**

This paper combines a Feature Pyramid Network with Pyramidal Detail Pooling to existing networks such as MRNet and ELNet to improve the detection of anterior cruciate ligament and meniscal lesions from MRIs. The proposed methodology is tested on two datasets and show improvements over the baseline approaches.

**Strengths:**

The approach uses Feature Pyramid Networks which are powerful at detecting small features.
The approach shows improvements over the baseline approaches.
The approach has potential to be plugged-in to other architectures  in support to other classification problems.

**Weaknesses:**

The work is strongly assuming that abnormalities are always located in the centre of an mri slice.
I suspect Radiologists would not be comfortable with using a system which does not investigate every part of the image (obviously lesions in the menisci should be searched in the menisci, or ligament lesions in the ligaments).

The manuscript has either very detailed descriptions of the mechanics or lacks details about the conducted experiments.
Contributions and limitations should be clearly stated.

The Related work section should not be in the appendix, because related works are very important. This applies with respect to the specific problem here solved as well as to the previous applications of Feature pyramid networks.

**Deanonymize Review:**

no

**Detailed Comments:**

Just to improve its impact, the abstract could be re-organised starting from what the work proposes.

It is true that the abnormalities are typically small features, however it is not true that they are only located in the centre of a slice. This claim is also sensible to the view.

I think there is some confusion when using the term MRI exam or slice making it difficult to completely comprehend part of the processing operation.

"An MRI volume is a sequence of S W × H MRI slices", in the methodology what does C refer to? MRI contrast? If so, does it mean that different views/contrast of the same MRI are inputted together?

While the authors claim the approach could be plugged in any diagnostic pipeline, this claim is recommended to be demonstrated in at least another task not knee related.

It is hard to follow the flow of operation done, the reader should not necessarily relate to the repository to replicate and apply the idea. Although releasing the code is not mandatory, by providing a link to the github repository, code availability becomes an expectation - and such expectation was not met.

Figure 1 and 2 could be supported by tables which describe what convolution were employed.

Namiri et al 2020 showed that a multi-stage approach performs better than end-to-end in ACL severity staging. It would be interesting to investigate if a multistage approach which involved detection of the region of interest followed by classification using the authors' proposed module would further boost performance.

Only the graph in Figure 4 gives indication that multiple networks, one per view were trained. It would be very useful to have a clear experimental section in the manuscript which clarifies what and how experiments were conducted


**Final Rating Justification:**

After the rebuttal I am incline to accept this paper, and it would be interesting to see it presented as a poster.
I still think the authors should update the manuscript, share the code as promised, improve the related work section and compare at least conceptually against similar works. This cannot be left for the future work.

**Justification Of The Preliminary Rating:**

This work investigates a very important and challenging task. However, at this stage, big portions of the paper are difficult to interpret. It is difficult to discern what is novel and what is just a re-use of previous ideas. Without a larger number of validating experiments it is hard to really appreciate the potential of the method.

**Paper Type:**

both

**Questions To Address In The Rebuttal:**

The authors are encouraged to address all the comments stated above.

In addition authors should show examples where the approach failed and compare these cases with the baseline counterparts, as well as the vice-versa.

Authors should comment about the claim that the approach could improve MRI interpretation for less specialized radiologists.

**Special Issue:**

no

---

> ### Author Response · Authors · 2021-03-17
> **Replies to AnonReviewer2 - Part 1**
>
> The authors would like to thank the reviewer for his/her extensive comments and suggestions.
>
> > The work is strongly assuming that abnormalities are always located in the centre of an mri slice. I suspect Radiologists would not be comfortable with using a system which does not investigate every part of the image (obviously lesions in the menisci should be searched in the menisci, or ligament lesions in the ligaments).
>
> Our proposed strategy exploits the idea of augmenting CNNs with prior information about the disorders in order to improve the diagnostic performance. Our proposed PDP module, which is designed after an analysis of the literature  (Hash, 2013; Nguyen et al., 2014; Stajduhar et al., 2017; Lecouvet et al., 2018; Bien et al., 2018), has to be considered as a generic baseline strategy to implement such a prior. We made this point more explicit in the revised version of the paper. Moreover, we remark that PDP does not pool just a single RoI, but multiple RoIs that increasingly focus on the image center. In this way, if a lesion is out of view of an inner RoI it can be still captured by an outer RoI of a previous level, which analyzes a larger RoI. At the lowest detail level, the RoI matches the size of the backbone's feature maps. The overall procedure performed allows to look for more detailed features, while guaranteeing that the abnormality is always captured by at least one detail level. Hence, the PDP module analyzes every part of the image, just with different levels of detail.
>
> > The manuscript has either very detailed descriptions of the mechanics or lacks details about the conducted experiments. Contributions and limitations should be clearly stated.
>
> We would like to remark that our strategy is applied to the pipelines of MRNet and ELNet without modifications to the original experimental settings, and from which we inherit the experimental settings. We added a recap of such in the appendix B of the revised paper, and for further details, we make the reader refer directly to their original papers. Contributions and limitations are better highlighted in the revised version.
>
> > The Related work section should not be in the appendix, because related works are very important. This applies with respect to the specific problem here solved as well as to the previous applications of Feature pyramid networks.
>
> The time allowed for the rebuttal was not sufficient for a complete reorganisation of the paper to include the related work in the main paper. However, we will follow the reviewer's suggestion and revise such a section for the camera ready version.
>
> > Just to improve its impact, the abstract could be re-organised starting from what the work proposes.
>
> The abstract has been revised to give a better presentation of the contribution.
>
> > It is true that the abnormalities are typically small features, however it is not true that they are only located in the centre of a slice. This claim is also sensible to the view.
>
> We agree with the reviewer that it is not always the case for abnormalities to be located in the centre of a slice, and we rephrased our statements to avoid confusion. However, the results achieved by our proposed solution demonstrate that the intuition of exploiting the information contained in particular slice areas (which comes from the analysis of the literature) leads to the improvement of disorder detection, even for different MRI views (as shown in Figure 4).
>
> > "An MRI volume is a sequence of S W × H MRI slices", in the methodology what does C refer to? MRI contrast? If so, does it mean that different views/contrast of the same MRI are inputted together?
>
> The $C$ term refers to the number of image channels used to represent an MRI slice. Typically, $C = 1$ as only gray-scale intensity levels are used. However, for the application of MRPyrNet to MRNet we used $C = 3$ (each slice is replicated three times) as in the original solution. This is done for the exploitation of the AlexNet backbone which has been pretrained with RGB images of the ImageNet dataset. This point has been made more clear in the revised version of the paper.
>
> > While the authors claim the approach could be plugged in any diagnostic pipeline, this claim is recommended to be demonstrated in at least another task not knee related.
>
> About the claim reported by the reviewer, we would like to point out that our proposed idea of combining an FPN and particular pooling strategies is general enough that can be of interest for other CNN-based diagnostic pipelines. However, we realize that our statement can lead to confused conclusions. Therefore, in the revised paper, such a statement has been rephrased.

---

> ### Author Response · Authors · 2021-03-17
> **Replies to AnonReviewer2 - Part 2**
>
> > It is hard to follow the flow of operation done, the reader should not necessarily relate to the repository to replicate and apply the idea. Although releasing the code is not mandatory, by providing a link to the github repository, code availability becomes an expectation - and such expectation was not met.
>
> The time allowed for the rebuttal was not sufficient for a complete reorganisation of the paper, but we will follow the reviewer's comment on the simplification of the methodology section for the camera ready version. We remark that the methodology described in the paper will be accompanied by the code. And the code, as usual, will be released only upon the acceptance of the paper for intellectual property purposes.
>
> > Figure 1 and 2 could be supported by tables which describe what convolution were employed.
>
> Thanks for the suggestion. A table reporting the details about the layers of the FPN has been introduced in the appendix B.
>
> > Namiri et al 2020 showed that a multi-stage approach performs better than end-to-end in ACL severity staging. It would be interesting to investigate if a multistage approach which involved detection of the region of interest followed by classification using the authors' proposed module would further boost performance.
>
> We thank the reviewer for suggesting this point. We agree that such an idea could be beneficial for our solution, and an investigation will be carried in future work.
>
> > Only the graph in Figure 4 gives indication that multiple networks, one per view were trained. It would be very useful to have a clear experimental section in the manuscript which clarifies what and how experiments were conducted
>
> The results presented in Figure 4 refer to the single-view models as proposed by Bien et al., 2018. These are first trained separately for the task of interest, and then their predictions are combined by a logistic regression to obtain a single probability estimate of disorder presence/absence. Additional details on these experimental settings have been added in the Appendix B of the revised paper.
>
> > In addition authors should show examples where the approach failed and compare these cases with the baseline counterparts, as well as the vice-versa.
>
> The results reported in Tables 1 and 2 show that MRPyrNet decreases the specificity of the baseline model. This is a limitation of the proposed solution and suggests that MRPyrNet is not able to capture relevant information in the absence of abnormalities. However, this is consistent with our goal, which is to extract more relevant information about the abnormalities, and this can be effectively done only when abnormalities are present. We discussed this and other points in the newly introduced section "Limitations" of the appendix.
>
> > Authors should comment about the claim that the approach could improve MRI interpretation for less specialized radiologists.
>
> The claim the reviewer refers was reported to point out the possible applications of deep learning models for knee disorder diagnosis in clinical workflows,  and it was initially discussed by Bien et al., 2018 and Tsai et al., 2020. However, we believe our solution has the potential to be implemented for radiologists aid in MRI interpretation. Indeed, supported by the ML community's achievements in Explainable AI, we think our approach could deliver a more precise interpretation in combination with techniques like GradCAM or others. Intuitively, such methods could highlight not only the slice area but also which RoI had more impact on the diagnostic decision and therefore guide radiologists in focusing on increasingly precise areas starting from the full slice area. Our team is currently investigating this idea.

---

### Meta-Review · Area_Chair1 · 2021-03-27

**Recommendation:** Accept (Poster)

**Metareview:**

The authors propose a new architecture to address a specific detection problem.

Since the architecture rely on existing components, the methodological novelty is marginal, however:

- the authors address a novel application
- they provide a thorough evaluation (with ablation study and statistical test)
- the authors took the reviewer’s remarks into account, and performed additional experiments
- their code will be made available.

Thus  I recommend acceptance of this paper as a poster.

**Paper Type:**

validation/application paper

---

### Decision · Program_Chairs · 2021-03-31

Accept